# Prevalence and outcomes of co-infection and superinfection with SARS-CoV-2 and other pathogens: A systematic review and meta-analysis

Jackson S. Musuuza[1,2], Lauren Watson[1], Vishala Parmasad[1], Nathan Putman-Buehler[1], Leslie Christensen[3], Nasia Safdar[1,2]*

**1** Division of Infectious Disease, Department of Medicine, University of Wisconsin School of Medicine and Public Health, Madison, WI, United States of America, **2** William S. Middleton Memorial Veterans Hospital, Madison, WI, United States of America, **3** Ebling Library for the Health Sciences, University of Wisconsin School of Medicine and Public Health, Madison, WI, United States of America

\* ns2@medicine.wisc.edu

**Data Availability Statement:** All relevant data are within the paper and its Supporting Information files.

## Abstract

### Introduction

The recovery of other pathogens in patients with SARS-CoV-2 infection has been reported, either at the time of a SARS-CoV-2 infection diagnosis (co-infection) or subsequently (superinfection). However, data on the prevalence, microbiology, and outcomes of co-infection and superinfection are limited. The purpose of this study was to examine the occurrence of co-infections and superinfections and their outcomes among patients with SARS-CoV-2 infection.

### Patients and methods

We searched literature databases for studies published from October 1, 2019, through February 8, 2021. We included studies that reported clinical features and outcomes of co-infection or superinfection of SARS-CoV-2 and other pathogens in hospitalized and non-hospitalized patients. We followed PRISMA guidelines, and we registered the protocol with PROSPERO as: CRD42020189763.

### Results

Of 6639 articles screened, 118 were included in the random effects meta-analysis. The pooled prevalence of co-infection was 19% (95% confidence interval [CI]: 14%-25%, $I^2$ = 98%) and that of superinfection was 24% (95% CI: 19%-30%). Pooled prevalence of pathogen type stratified by co- or superinfection were: viral co-infections, 10% (95% CI: 6%-14%); viral superinfections, 4% (95% CI: 0%-10%); bacterial co-infections, 8% (95% CI: 5%-11%); bacterial superinfections, 20% (95% CI: 13%-28%); fungal co-infections, 4% (95% CI: 2%-7%); and fungal superinfections, 8% (95% CI: 4%-13%). Patients with a co-infection or superinfection had higher odds of dying than those who only had SARS-CoV-2 infection (odds ratio = 3.31, 95% CI: 1.82–5.99). Compared to those with co-infections, patients with

**Funding:** NS received research support for this work from the National Institute of Allergy and Infectious Diseases of the National Institutes of Health under Award Number DP2AI144244. The content is solely the responsibility of the authors and does not necessarily represent the official views of the National Institutes of Health. The funding agency did not play any role in the study's design, data collection, analysis, decision to publish or preparation of the manuscript.

**Competing interests:** The authors have declared that no competing interests exist.

superinfections had a higher prevalence of mechanical ventilation (45% [95% CI: 33%-58%] vs. 10% [95% CI: 5%-16%]), but patients with co-infections had a greater average length of hospital stay than those with superinfections (mean = 29.0 days, standard deviation [SD] = 6.7 vs. mean = 16 days, SD = 6.2, respectively).

## Conclusions

Our study showed that as many as 19% of patients with COVID-19 have co-infections and 24% have superinfections. The presence of either co-infection or superinfection was associated with poor outcomes, including increased mortality. Our findings support the need for diagnostic testing to identify and treat co-occurring respiratory infections among patients with SARS-CoV-2 infection.

## Introduction

The coronavirus disease 2019 (COVID-19) pandemic is associated with high morbidity and mortality [1, 2]. Current evidence shows that severe acute respiratory syndrome coronavirus 2 (SARS-CoV-2), the causative agent of COVID-19, is primarily transmitted through respiratory droplets [3, 4] from symptomatic, asymptomatic, or pre-symptomatic individuals [4, 5]. Similar to other respiratory pathogens, such as influenza, where approximately 25% of older patients get secondary bacterial infections [6, 7], both superinfections and co-infections with SARS-CoV-2 have been reported [8–10]. However, there is scarce data on the frequency of co-infection and superinfections by viral, bacterial, or fungal infections and associated clinical outcomes among patients infected with SARS-CoV-2 [8–10].

We define co-infection as the recovery of other respiratory pathogens in patients with SARS-CoV-2 infection at the time of a SARS-CoV-2 infection diagnosis and superinfection as the subsequent recovery of other respiratory pathogens during care for SARS-CoV-2 infection. Two previous reviews have examined the prevalence of bacterial and fungal co-infection or superinfection in SARS-CoV-2 infected patients [11, 12]. In addition, prior work suggests outcome differences in patients with co-infections vs. superinfections. For example, Garcia-Vidal et al., showed that SARS-CoV-2 infected patients with superinfection s had a longer length of hospital stay (LOS) and higher mortality, while those with co-infections had a higher frequency of admission to the ICU [13].

Diagnostic testing and therapeutic decision-making may be affected by the presence of co-infection or superinfection with SARS-CoV-2 and other respiratory pathogens.

Therefore, we conducted a systematic review and meta-analysis to examine the occurrence and outcomes (e.g., LOS) of respiratory co-infections and superinfections among patients infected with SARS-CoV-2.

## Materials and methods

We conducted this systematic review in accordance with the Preferred Reporting in Systematic Reviews and Meta-Analyses (PRISMA) guidelines [14]. We registered this review with PROSPERO: CRD42020189763 [15]. The protocol is available as a S1 File.

### Data sources and searches

With the help of a health sciences librarian (LC), we searched PubMed, Scopus, Wiley, Cochrane Central Register of Controlled Trials, Web of Science Core Collection, and CINAHL

Plus databases to identify English-language studies published from October 1, 2019, through February 8, 2021. We executed the search in PubMed and translated the keywords and controlled vocabulary for the other databases, and additional articles were added from reference lists of pertinent articles. The following keywords were used for the search: "coronavirus","coronavirus infections", "HCoV", "nCoV", "Covid", "SARS", "COVID-19", "2019 nCoV", "nCoV 19", "SARS-CoV-2", "SARS coronavirus2", "2019 novel corona virus", "Human", "pneumonia", "influenza", "severe acute respiratory syndrome", "co-infection", "Superinfection", "bacteria", "fungus", "concomitant", "pneumovirinae", "pneumovirus infections", "respiratory syncytial viruses", "metapneumovirus", "influenza", "human", "respiratory virus", "bacterial Infections", "viral infection", "fungal infection", "upper respiratory", "oxygen inhalation therapy", "intensive care units", "nursing homes", "subacute care", "skilled nursing", "intermediate care", "patient discharge", "mortality", "morbidity" and English filter. A complete description of our search strategy is available as a S2 File.

## Study selection

Citations were uploaded into Covidence®, an online systematic review software for the study selection process. Two authors (JSM and LW) independently screened titles and abstracts and read the full texts to assess if they met the inclusion criteria. The authors met and discussed any articles where there was conflict and decided to either include or exclude such articles. Inclusion criteria were randomized clinical trials (RCTs), quasi-experimental and observational human studies that reported clinical features and outcomes of co-infection or superinfection of SARS-CoV-2 (laboratory-confirmed) and other pathogens–fungal, bacterial, or other viruses–in hospitalized and non-hospitalized patients. We excluded studies that did not report co-infection or superinfection, editorials, reviews, qualitative studies, those published in a non-English language, articles where full texts were not available, and non-peer-reviewed preprints.

## Data extraction

Three reviewers (JSM, LW, and VP) independently abstracted data from individual studies using a standardized template. We abstracted data on study design/methodology, location and setting (intensive care unit [ICU], inpatient non-ICU, or outpatient, where applicable), study population, use of antibiotics, proportion of patients with co-infections, implicated pathogens, method of detection of co-infections and superinfections (laboratory-verified or clinical features only), type of infection (bacterial, viral, or fungal), and outcomes of co-infected patients (death, mechanical ventilation, discharge disposition, length of hospital stay, or mild illness). Discrepancies were resolved by discussion between the three abstractors.

## Risk of bias assessment

Risk of bias assessment was conducted by three authors (JSM, LW, and VP) independently. We used two study quality assessment tools, one specific to case series [16], and one for non-case series study designs [17].

The tool for case series examines four domains: selection, ascertainment, causality, and reporting [16]. The selection domain helps to assess whether participants included in a study are representative of the entire population from which they arise. Ascertainment assesses whether the exposure and outcome were adequately ascertained. Causality assesses the potential for alternative explanations and specifically for our study whether the follow-up was long enough for outcomes to occur. Reporting evaluates if a study described participants in sufficient detail to allow for replication of the findings. This tool consists of eight items, but only

five were applicable to our study [16]. When an item was present in a study, a score of 1 was assigned and 0 if the item was missing. We added the scores (minimum of 0 and a maximum of 5) and assigned the risk of bias as follows: low risk (5), medium risk (3–4), high risk (0–2).

For non-case series studies, we used the Modified Downs and Black risk assessment scale to assess the quality of cohort studies and RCTs [17]. This scale consists of 27 items that assess study characteristics, such as internal validity (bias and confounding), statistical power, and external validity. We scored studies as low risk (score 20–27, medium risk (score 15–19), or high risk (score ≤14).

### Data synthesis and analysis

The primary outcome was the prevalence of co-infections or superinfections by viral, bacterial, or fungal respiratory infections and SARS-CoV-2. We examined whether co-infection or superinfection was associated with an increased risk for the following patient outcomes: 1) mechanical ventilation, 2) admission to the ICU, 3) mortality and LOS.

We estimated the proportion of patients with co-infection or superinfection of viral, bacterial, and fungal respiratory infections and SARS-CoV-2. We anticipated a high level of heterogeneity given the novelty of COVID-19 and potential differences in testing and management of COVID-19 in the healthcare systems of the countries where the studies were conducted. We conducted all statistical analyses using Stata software, version 16.0 (Stata Corp. College Station, Texas). We used the "metan" and "metaprop" commands in Stata to estimate the pooled proportion of co-infection and superinfection and COVID-19 using a random effects model (Der-Simonian Laird) [18, 19]. We stabilized the variance using the Freeman-Tukey arcsine transformation methodology in order to correctly estimate extreme proportions (i.e., those close to 0% or 100%) [18]. We assessed heterogeneity using the $I^2$ statistic. Frequencies of outcome variables and study characteristics were estimated using descriptive statistics. For example, in studies where data on co-infecting or super-infecting pathogens were reported, we extracted and tallied the number of different pathogens reported. We calculated the proportion of pathogens using the number of pathogens as the numerator and the total number of pathogens of each type (bacteria, viruses, and fungi) from all the studies as the denominator.

We did not assess for publication bias because standard methods, such as funnel plots and associated tests, were developed for comparative studies and therefore do not produce reliable results for meta-analysis of proportions [20, 21].

## Results

Our search yielded 14457 records; we excluded 7818 duplicates and screened 6639 articles. At the abstract and title review stage, we excluded 6273 articles, leaving 366 articles for full-text review. Of these, 118 articles met the inclusion criteria and were included in this meta-analysis. The most frequent reason for exclusion of studies at the full-text review stage was the absence of superinfection or co-infection data (Fig 1).

Approximately half of the studies (60/118) were retrospective cohort studies, 35% (42/118) were cases series, and 9% (11/118) were prospective cohort studies. There were two case-control studies, two cross-sectional studies, and one clinical trial. The majority of the studies were conducted in China (42% [49/118)]) and the US (15% [18/118]). Most of the studies were conducted in a mixed setting (i.e., ICU and non-ICU setting; 72% [85/118]) and 92% (108/118) were conducted exclusively in hospitalized patients. The majority of studies were conducted among adults (73% [86/118]). Sixty-seven (57%) of the included studies reported that patients included had co-infections, 37% (44/118) reported superinfections, and 6% (7/118) reported both co-infections and superinfections among patients. Viral co-infections in patients were

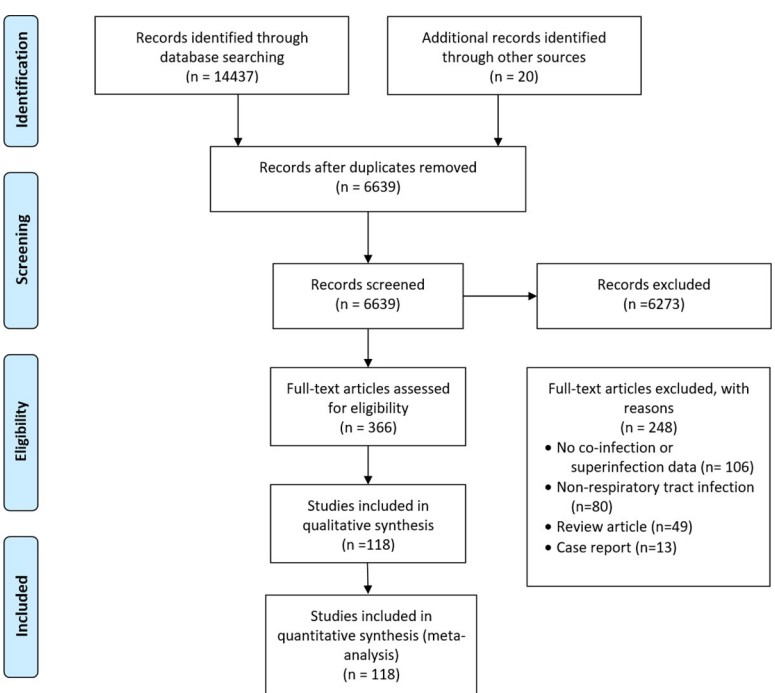

**Fig 1. Study selection flow diagram: Adapted from the PRISMA guideline [11].**

reported in 67% (55/81) of the studies, bacterial infections in 74% (78/105), fungal in 48% (35/73) of studies. Not all of the 118 studies reported data on viral, bacterial or fungal infections (Table 1). Seventy percent (83/118) of the studies reported data on antibiotic use. Of these, antibiotics were administered in 98% (81/83) of the studies.

The pooled prevalence of co-infection was 19% (95% confidence interval [CI]: 14%-25%; $I^2$ = 98%). The highest prevalence of co-infection was observed among non-ICU patients at 29% (95% CI: 14%-46%), while it was 18% (95% CI: 12%-25%) among combined ICU and non-ICU patients, and 16% (95% CI: 8%-25%) among only ICU co-infected patients (Fig 2). The pooled prevalence of superinfection was 24% (95% CI: 19%-30%), with the highest prevalence among ICU patients (41% [95% CI: 24%-58%]) (Fig 3).

Pooled prevalence of pathogen type stratified by co- or superinfection was: viral co-infections, 10% (95% CI: 6%-14%) and viral superinfections, 4% (95% CI: 0%-10%); bacterial co-infections, 8% (95% CI: 5%-11%) and bacterial superinfections, 20% (95% CI: 13%-28%); and fungal co-infections, 4% (95% CI: 2%-7%) and fungal superinfections, 8% (95% CI: 4%-13%) (S1–S3 Figs).

Seventy-eight studies reported data on specific organisms associated with co-infection or superinfection in COVID-19 patients (Table 2). Among patients with co-infections, the three most frequently identified bacteria were *Klebsiella pneumoniae* (9.9%), *Streptococcus pneumoniae* (8.2%), and *Staphylococcus aureus* (7.7%). The three most frequently identified viruses among co-infected patients were influenza type A (22.3%), influenza type B (3.8%), and respiratory syncytial virus (3.8%). For fungi, *Aspergillus* was the most frequently reported among those co-infected.

Among those with superinfections, the three most frequently identified bacteria were *Acinetobacter spp*. (22.0%), *Pseudomonas* (10.8%), and *Escherichia coli* (6.9%). For viruses, Rhinovirus was the most frequently identified among those with superinfections, and for fungi, *Candida sp*. was the most frequent (18.8%).

**Table 1. Main characteristics of included studies.**

| Study | Study design | Country | Setting | Number of patients | Age group of patients | Gender (% male) | ICU (%) | Patients who were ventilated n (%) | Patients who died n (%) | Viral co-infections n (%) | Bacterial co-infection n (%) | Fungal co-infections n (%) | Risk of bias |
|---|---|---|---|---|---|---|---|---|---|---|---|---|---|
| Arentz, 2020 [22] | Case series | USA | ICU[a] | 21 | Adults | 52 | 100 | 15 (71) | 11 (52) | 3 (14) | 1 (50) | 0 (0) | Medium |
| Barrasa, 2020 [23] | Case series | Spain | ICU | 48 | Adults | 56 | 100 | 45 (94) | 16 (33) | 0 (0) | 6 (13) | 0 (0) | Low |
| Campochiaro, 2020 [24] | Prospective cohort | Italy | ICU and non-ICU | 65 | Adults | 29 | 6 | 25 (38) | 16 (25) | 0 (0) | 1 (2) | 0 (0) | Low |
| Chen, 2020 [25] | Case series | China | ICU | 99 | Adults | 68 | 100 | 17 (17) | 11 (11) | 0 (0) | 1 (1) | 4 (4) | Medium |
| Cuadrado-Payán, 2020 [26] | Case series | Spain | ICU | 4 | Adults | 75 | 75 | 3 (75) | 0 (0) | 4 (100) | 0 (0) | 0 (0) | High |
| Ding, 2020 [27] | Case series | China | Non-ICU | 115 | Adults | NR[b] | 0 | 0 (0) | 0 (0) | 5 (4) | 0 (0) | 0 (0) | Medium |
| Dong, 2020 [28] | Case series | China | Non-ICU | 11 | Adults/children | 54 | 0 | 1 (9) | 0 (0) | 1 (9) | 0 (0) | 0 (0) | Medium |
| Du, 2020 [29] | Case series | China | ICU | 109 | Adults | 67.9 | 48.6 | 33 (30) | 109 (100) | 0 (0) | NR | NR | Low |
| Fan, 2020 [30] | Retrospective cohort | China | ICU and non-ICU | 50 | Adults | 83 | 54 | 23 (46) | 12 (24) | 0 (0) | 5 (10) | 5 (10) | Low |
| Feng, 2020 [31] | Case series | China | ICU and non-ICU | 476 | Adults | 56.9 | 26 | 70 (15) | 38 (8) | 0 (0) | 35 (7) | 0 (0) | Medium |
| Garazzino, 2020 [32] | Retrospective cohort | Italy | ICU and non-ICU | 168 | Children | 55.9 | 1.1 | 2 (1) | 0 (0) | 10 (6) | 1 (0.5) | 0 (0) | Low |
| Gayam, 2020 [33] | Case series | USA | ICU and non-ICU | 350 | Adults | 33 | NR | NR | NR | 0 (0) | 1 (0.3) | 0 (0) | Medium |
| Huang, 2020 [34] | Case series | China | ICU and non-ICU | 41 | Adults | 73 | 32 | 4 (10) | 6 (15) | 0 (0) | 1 (2) | 0 (0) | Medium |
| Kakuya, 2020 [35] | Case series | Japan | Non-ICU | 3 | Children | 100 | 0 (0) | 0 (0) | 0 (0) | 1 (33) | 0 (0) | 0 (0) | Low |
| Khodamoradi, 2020 [36] | Case series | Iran | Non-ICU | 4 | Adults | 75 | 0 | 0 (0) | 0 (0) | 4 (100) | 0 (0) | 0 (0) | Medium |
| Kim, 2020 [37] | Retrospective cohort | USA | Non-ICU | 115 | Adults/children | 45 | 0 | 0 (0) | 0 (0) | 25 (22) | 0 (0) | 0 (0) | Low |
| Koehler, 2020 [38] | Case series | Germany | ICU | 19 | Adults | NR | 100 | NR | 3 (16) | 2 (11) | 0 (0) | 5 (26) | Medium |
| Lian, 2020 [39] | Retrospective cohort | China | ICU and non-ICU | 788 | Children/Adults | 52 | 3 | 18 (2) | 0 (0) | NR | 0 (0) | 0 (0) | Low |
| Lin, 2020 [8] | Case series | China | ICU and non-ICU | 92 | Adults | NR | NR | NR | NR | 6 (7) | NR | NR | Medium |
| Liu, 2020 [40] | Retrospective cohort | China | ICU and non-ICU | 12 | Children/Adults | 66 | NR | 6 (50) | NR | 0 (0) | 2 (17) | 0 (0) | Low |
| Lv, 2020 [41] | Retrospective cohort | China | ICU and non-ICU | 354 | Adults | 49 | NR | NR | 11 (3) | 1 (0.3) | 32 (9) | 6 (2) | Low |
| Ma, 2020 [42] | Retrospective cohort | China | NR | 93 | Adults | 55 | NR | NR | 44 (47) | 46 (49) | 0 (0) | 0 (0) | Low |
| Mannheim, 2020 [43] | Case series | USA | ICU and non-ICU | 64 | Children | 56 | 11 | NR | 0 (0) | 3 (5) | 1 (2) | 0 (0) | Medium |
| Mo, 2020 [44] | Case series | China | ICU and non-ICU | 155 | Adults | 55 | NR | 36 (23) | 22 (14) | 13 (8) | 2 (1) | 0 (0) | Medium |
| Nowak, 2020 [9] | Case series | USA | ICU and non-ICU | 1204 | Adults | 56 | NR | NR | NR | 36 (3) | 0 (0) | 0 (0) | Medium |
| Ozaras, 2020 [45] | Case series | Turkey | ICU and non-ICU | 1103 | Adults | 50 | NR | NR | NR | 6 (0.5) | 0 (0) | 0 (0) | Medium |
| Palmieri, 2020 [46] | Retrospective cohort | Italy | ICU and non-ICU | 3032 | Children/Adults | 67 | NR | NR | 3032 (100) | NR | NR | NR | Low |

(*Continued*)

**Table 1.** (Continued)

| Study | Study design | Country | Setting | Number of patients | Age group of patients | Gender (% male) | ICU (%) | Patients who were ventilated n (%) | Patients who died n (%) | Viral co-infections n (%) | Bacterial co-infection n (%) | Fungal co-infections n (%) | Risk of bias |
|---|---|---|---|---|---|---|---|---|---|---|---|---|---|
| Peng, 2020 [47] | Retrospective cohort | China | ICU and non-ICU | 75 | Children | 58 | NR | NR | 0 (0) | 8 (11) | 31 (41) | 0 (0) | Low |
| Pongpirul, 2020 [48] | Case series | Thailand | ICU and non-ICU | 11 | Adults | 54 | NR | 0 (0) | 0 (0) | 2 (18) | 5 (45) | 0 (0) | Low |
| Richardson, 2020 [49] | Case series | USA | ICU and non-ICU | 5700 | Adults | 60 | 14.2 | 1151 (20) | 553 (10) | 39 (0.7) | 3 (0.1) | 0 (0) | Low |
| Sun, 2020 [50] | Retrospective cohort | China | ICU and non-ICU | 36 | Children | 61 | NR | NR | 1 (3) | 1 (3) | 1 (3) | 0 (0) | Medium |
| Tagarro, 2020 [51] | Retrospective cohort | Spain | ICU and non-ICU | 41 | Children | 44 | 9.7 | 4 (10) | 0 (0) | 2 (5) | 0 (0) | 0 (0) | Low |
| Wan, 2020 [52] | Case series | China | ICU and non-ICU | 135 | Adults | 53 | NR | 28 (21) | 1 (0.7) | NR | NR | NR | Medium |
| Wang Y, 2020 [53] | Case series | China | ICU and non-ICU | 55 | Adults | 40 | 0 | 0 (0) | 0 (0) | 1 (2) | 1 (2) | 1 (3) | Low |
| Wang L, 2020 [54] | Case series | China | ICU and non-ICU | 339 | Adults | 49 | NR | NR | 65 (19) | 0 (0) | 1 (0.3) | 1 (0.3) | Low |
| Wang R, 2020 [55] | Case series | China | ICU and non-ICU | 125 | Adults | 56.8 | 15.2 | 4 | 0 (0) | 1 (0.8) | 9 (7) | 9 (7) | Medium |
| Wang Y, 2020 [56] | Clinical trial | China | ICU and non-ICU | 237 | Adults | 56 | NR | 21 (9) | 14 (6) | NR | NR | NR | Medium |
| Wee, 2020 [57] | Prospective cohort | Singapore | ICU and non-ICU | 3807 | Adults | NR | NR | NR | 1 (0.02) | 3 (0.08) | NR | NR | Medium |
| Wu C, 2020 [58] | Retrospective cohort | China | ICU and non-ICU | 201 | Adults | 63.7 | 26.4 | 67 (33) | 44 (22) | 1 (0.5) | 0 (0) | 0 (0) | Low |
| Xia, 2020 [59] | Case series | China | ICU and non-ICU | 20 | pediatric | 65 | NR | 0 (0) | 0 (0) | 4 (0.2) | 1 (5) | 1 (5) | Medium |
| Yang X, 2020 [60] | Case series | China | ICU | 710 | Adults | 67 | 100 | 37 (5) | 32 (4) | 0 (0) | 4 (0.6) | 4 (0.6) | Low |
| Yi, 2020 [61] | Case series | USA | ICU and non-ICU | 132 | Adult | 62 | 50 | 5 (4) | 1 (0.8) | NR | NR | NR | Medium |
| Zhang J, 2020 [62] | Case series | China | ICU and non-ICU | 140 | Adults | 50.7 | NR | NR | NR | 2 (1) | 1 (0.7) | 1 (0.7) | Medium |
| Zhang G, 2020 [63] | Case series | China | ICU and non-ICU | 221 | Adults | 48.9 | 80 | 26 (12) | 5 (2) | 2 (0.9) | 6 (3) | 6 (3) | Medium |
| Zhao, 2020 [64] | Case series | China | ICU and non-ICU | 34 | Adults | 57.9 | 0 | 0 (0) | 0 (0) | 1 (3) | 1 (3) | 0 (0) | Medium |
| Zheng, 2020 [65] | Case series | China | ICU and non-ICU | 1001 | Adult and pediatric | NR | NR | NR | NR | 2 (0.2) | NR | NR | Low |
| Zhou, 2020 [66] | Retrospective cohort | China | ICU and non-ICU | 191 | Adult | 62 | 26 | 32 (17) | 54 (28) | NR | NR | NR | Low |
| Zhu, 2020 [67] | Retrospective cohort | China | ICU and non-ICU | 257 | Adult and pediatric | 53.7 | 1.16 | 0 (0) | 0 (0) | 9 (3) | 11 (4) | 11 (4) | Low |
| Alvares P, 2020 [68] | Retrospective cohort | Brazil | ICU and non-ICU | 32 | Pediatric | 59.3 | 9.3 | 2 (6) | 1 (3) | 1 (3) | 1 (3) | NR | Medium |
| Borman, 2020 [69] | Case series | UK | ICU | 719 | Adults | NR | 100.0 | NR | NR | NR | NR | 3NR | Low |
| Chaudhary W, 2020 [70] | Case series | Brunei Darussalam | ICU and non-ICU | 141 | Adults | NR | | NR | NR | NR | 7 (5) | NR | Low |
| Cheng L, 2020 [71] | Retrospective cohort | Hong Kong | ICU and non-ICU | 147 | Adults | 85.0 | 3.0 | NR | NR | NR | 4 (3) | NR | Low |

(Continued)

**Table 1.** (Continued)

| Study | Study design | Country | Setting | Number of patients | Age group of patients | Gender (% male) | ICU (%) | Patients who were ventilated n (%) | Patients who died n (%) | Viral co-infections n (%) | Bacterial co-infection n (%) | Fungal co-infections n (%) | Risk of bias |
|---|---|---|---|---|---|---|---|---|---|---|---|---|---|
| Cheng Y, 2020 [72] | Retrospective cohort | China | ICU and non-ICU | 213 | Adults | 50.2 | | 2 (1) | 8 (4) | 97 (46) | NR | NR | Low |
| Cheng K, 2020 [73] | Retrospective cohort | China | NR | 212 | Adults/Children | 51.0 | | 19 (9) | NR | NR | 13 (6) | NR | Low |
| Contou D, 2020 [74] | Retrospective cohort | France | ICU | 92 | Adults | 79.0 | 100.0 | 83 (90) | 45 (49) | NR | 32 (35) | NR | Low |
| Dupont D, 2020 [75] | Case series | France | ICU | 19 | Adults | 78.0 | 100.0 | 18 (95) | NR | NR | NR | 19 (100) | Low |
| Elabbadi A, 2020 [76] | Case series | France | ICU | 101 | Adults | 78.2 | 100.0 | 83 (82) | 21 (21) | NR | 10 (10) | NR | Low |
| Falces-Romero, 2020 [77] | Retrospective cohort | Spain | ICU and non-ICU | 10 | Adults | 80.0 | 70.0 | 7 (70) | 7 (70) | NR | 0 | 10 (100) | Medium |
| Falcone M, 2020 [78] | Prospective cohort | Italy | ICU and non-ICU | 315 | Adults | 66.6 | 26.9 | 55 (17) | 70 (22) | NR | 11 (3) | 2 (1) | Medium |
| Fu Y, 2020 [79] | Case series | China | ICU and non-ICU | 5 | Adults | 80.0 | 100.0 | 5 (100) | NR | NR | 5 (100) | 2 (40) | Low |
| Garcia-Menino, 2021 [80] | Case series | Spain | ICU | 7 | Adults | 86.0 | 100.0 | NR | 1 (14) | NR | 7 (100) | NR | Low |
| Garcia-Vidal, 2021 [81] | Prospective cohort | Spain | ICU and non-ICU | 989 | Adults | 55.8 | 15.0 | NR | 103 (10) | 6 (1) | 47 (5) | 7 (1) | Low |
| Gouzien, 2020 [82] | Retrospective cohort | France | ICU | 53 | Adults | 67.9 | 100.0 | 53 (100) | 39 (74) | NR | NR | 1 (2) | Medium |
| Hashemi S, 2020 [83] | Case series | Iran | ICU and non-ICU | 105 | Adults/Children | NR | | NR | 105 (100) | NR | NR | NR | Low |
| Hazra A, 2020 [84] | Retrospective cohort | USA | ICU and non-ICU | 459 | NR | NR | | NR | NR | 6 (1) | NR | NR | High |
| He Bing, 2020 [85] | Retrospective cohort | China | NR | 21 | Adults/Children | NR | | NR | 0 | NR | 2 (10) | 4 (19) | Medium |
| Hirotsu Y, 2020 [86] | Prospective cohort | Japan | non-ICU | 191 | NR | NR | | NR | NR | 32 (17) | NR | NR | Medium |
| Hughes, 2020 [87] | Case series | UK | ICU | 836 | Adults | 62.0 | 31.0 | NR | 262 (31) | NR | 5 (1) | 27 (3) | Low |
| Karaba, 2020 [88] | Retrospective cohort | USA | ICU and non-ICU | 1016 | Adults | 54.0 | 12.0 | NR | NR | 2 NR | 1NR | NR | Low |
| Kolenda, 2020 [89] | Prospective cohort | France | ICU | 99 | NR | NR | 100.0 | NR | NR | NR | 17 (17) | NR | Low |
| Kumar, 2021 [90] | Retrospective cohort | USA | ICU and non-ICU | 1573 | Adults | 57.9 | 31.0 | 247 (16) | 413 (26) | NR | 48 (3) | 9 (1) | Low |
| Lardaro T, 2020 [91] | Retrospective cohort | USA | ICU and non-ICU | 542 | Adults | 49.6 | 15.9 | 159 (29) | 78 (14) | NR | 8 (1) | NR | Medium |
| Lehmann C, 2020 [92] | Retrospective cohort | USA | ICU and non-ICU | 321 | Adults | 48.0 | 5.0 | NR | 22 (7) | 5 (2) | 7 (2) | NR | Medium |
| Lendorf, 2020 [93] | Retrospective cohort | Denmark | ICU and non-ICU | 115 | Adults/Children | 60.0 | 18.0 | 12 (10) | 16 (14) | NR | 9 (8) | 1 (1) | Medium |
| Li J, 2020 [94] | Retrospective cohort | China | ICU and non-ICU | 102 | Adults/Children | 66.7 | | NR | 50 (49) | NR | 159 (156) | NR | Medium |
| Li Z, 2020 [95] | Retrospective cohort | China | ICU and non-ICU | 32 | Adults | 62.5 | 40.0 | 6 (19) | NR | 6 (19) | 10 (31) | 2 (6) | High |

(*Continued*)

**Table 1.** (Continued)

| Study | Study design | Country | Setting | Number of patients | Age group of patients | Gender (% male) | ICU (%) | Patients who were ventilated n (%) | Patients who died n (%) | Viral co-infections n (%) | Bacterial co-infection n (%) | Fungal co-infections n (%) | Risk of bias |
|---|---|---|---|---|---|---|---|---|---|---|---|---|---|
| Ma L, 2020 [96] | Retrospective cohort | China | ICU and non-ICU | 250 | Adults | 46.0 | | 5 (2) | 4 (2) | 4 (2) | 2 (1) | NR | Low |
| Mahmoudi H, 2020 [97] | Cross-sectional study | Iran | ICU and non-ICU | 342 | Adults | NR | | NR | NR | NR | 6 (2) | NR | Medium |
| Mendes N, 2020 [98] | Retrospective cohort | USA | ICU and non-ICU | 242 | Adults | 50.8 | | 54 (22) | 52 (21) | NR | 6 (2) | NR | Low |
| Mughal, 2020 [99] | Restrospective cohort | USA | ICU and non-ICU | 129 | Adults | 62.8 | 30.2 | 30 (23) | 20 (16) | NR | NR | NR | Low |
| Nasir N, 2020 [100] | Retrospective cohort | Pakistan | ICU and non-ICU | 30 | Adults | 83.0 | 33.0 | 24 (80) | 7 (23) | NR | 6 (20) | 7 (23) | Low |
| Nasir N, 2020 [101] | Retrospective cohort | Pakistan | ICU and non-ICU | 147 | Adults | 60.0 | | | NR | NR | 9 (6) | 1 (1) | Medium |
| Ng K F, 2020 [102] | Case series | China | ICU and non-ICU | 8 | Pediatric | 25.0 | 25.0 | NR | NR | 5 (63) | NR | NR | Low |
| Nori, 2021 [103] | Retrospective cohort | USA | ICU and non-ICU | 152 | Adults/Children | 59.0 | 55.9 | NR | 86 (57) | NR | 112 (74) | 3 (2) | Low |
| Obata, 2020 [104] | Retrospective cohort | USA | ICU and non-ICU | 226 | Adults | 55.1 | 24.8 | NR | 41 (18) | NR | 8 (4) | 8 (4) | Medium |
| Oliva, 2020 [105] | Case series | Italy | ICU and non-ICU | 7 | Adults | 57.0 | 14.3 | NR | NR | NR | 7 (100) | NR | Low |
| Papamanoli, 2020 [106] | Retrospective cohort | USA | ICU and non-ICU | 447 | Adults | 66.0 | 45.2 | 115 (26) | 102 (23) | NR | NR | NR | Low |
| Peci A, 2021 [107] | Case-control | Canada | ICU and non-ICU | 325 | Adults/Children | NR | | NR | NR | 8 (2) | NR | NR | Low |
| Pereira, 2021 [108] | Case-control | New York | ICU and non-ICU | 87 | Adults | 60.9 | 48.3 | NR | 32 (37) | 10 (11) | 6 (7) | 1 (1) | Medium |
| Pettit, 2020 [109] | Retrospective cohort | USA | ICU and non-ICU | 148 | Adults | 37.5 | 70.3 | 48 (32) | 46 (31) | 1 (1) | 14 (9) | 2 (1) | Low |
| Pickens, 2021 [110] | Retrospective cohort | Chicago | ICU | 179 | Adults | 61.5 | 100.0 | 179 (100) | 34 (19) | NR | 28 (16) | NR | Low |
| Ramadan H, 2021 [111] | Prospective cohort | Egypt | ICU and non-ICU | 260 | Adults | 55.4 | | 8 (3) | 24 (9) | NR | 37 (14) | NR | Low |
| Reig S, 2020 [112] | Retrospective cohort | Germany | ICU and non-ICU | 213 | Adults | 61.0 | 33.0 | 57 (27) | 51 (24) | NR | 26 (12) | 6 (3) | Low |
| Ripa M, 2020 [113] | Prospective cohort | Italy | ICU and non-ICU | 731 | Adults | 68.0 | 12.0 | NR | 194 (27) | NR | 24 (3) | 11 (2) | Low |
| Rothe K, 2020 [114] | Retrospective cohort | Germany | ICU and non-ICU | 140 | Adults | 64.0 | 15.0 | 41 (29) | NR | NR | NR | 9 (6) | Low |
| Segrelles-Calvo G, 2021 [115] | Case series | Spain | ICU and non-ICU | 7 | Adults | 71.0 | 86.0 | 7 (100) | 5 (71) | NR | NR | 7 (100) | Low |
| Sharifipour E, 2020 [116] | Prospective cohort | Iran | ICU | 19 | Adults | 58.0 | 100.0 | 19 (100) | 18 (95) | NR | 19 (100) | NR | Low |
| Sogaard, 2021 [117] | Retrospective cohort | Switzerland | ICU and non-ICU | 162 | Adults | 61.1 | 25.3 | NR | 17 (10) | 5 (3) | 19 (12) | 3 (2) | Low |
| Soriano, 2021 [118] | Retrospective cohort | Spain | ICU | 83 | Adults | 79.0 | 100.0 | 78 (94) | 20 (24) | NR | 7 (8) | NR | Low |
| Tang, 2021 [119] | Retrospective cohort | China | NR | 78 | Adults/Children | 53.0 | | NR | NR | 4 (5) | 5 (6) | NR | Low |

(Continued)

**Table 1.** (Continued)

| Study | Study design | Country | Setting | Number of patients | Age group of patients | Gender (% male) | ICU (%) | Patients who were ventilated n (%) | Patients who died n (%) | Viral co-infections n (%) | Bacterial co-infection n (%) | Fungal co-infections n (%) | Risk of bias |
|---|---|---|---|---|---|---|---|---|---|---|---|---|---|
| Torrego, 2020 [120] | Retrospective cohort | Spain | ICU | 163 | NR | NR | 100.0 | 139 (85) | 23 (14) | NR | 18 (11) | NR | High |
| Townsend, 2020 [121] | Prospective cohort | Ireland | ICU and non-ICU | 117 | Adults | 63.0 | 29.1 | NR | 17 (15) | NR | 6 (5) | 1 (1) | Low |
| Verroken, 2020 [122] | Prospective cohort | Belgium | ICU | 32 | NR | NR | 100.0 | NR | NR | NR | 13 (41) | NR | Medium |
| Wang L, 2020 [123] | Retrospective cohort | UK | ICU and non-ICU | 1396 | Adults | 65.0 | 30.0 | NR | 420 (30) | NR | 11 (1) | NR | Low |
| Wei L, 2020 [124] | Retrospective cohort | China | non-ICU | 43 | Adults | 0.0 | 0.0 | NR | NR | 15 (35) | NR | NR | Low |
| White P, 2020 [125] | Retrospective cohort | UK | ICU and non-ICU | 135 | Adults | 69.0 | | NR | 51 (38) | NR | NR | 36 (27) | Low |
| Wu Q, 2020 [126] | Retrospective cohort | China | NR | 74 | Pediatric | 59.5 | | 1 (1) | NR | 10 (14) | 16 (22) | NR | Low |
| Xia P, 2020 [127] | Retrospective cohort | China | ICU | 81 | Adults | 66.7 | 100.0 | 66 (81) | 60 (74) | NR | 34 (42) | NR | Low |
| Xu J, 2020 [128] | Retrospective cohort | China | ICU | 239 | Adults | 59.8 | 100.0 | 165 (69) | 147 (62) | NR | 25 (10) | NR | Low |
| Xu S, 2020 [129] | Retrospective cohort | China | ICU and non-ICU | 64 | Adults | 0.0 | 1.6 | NR | NR | 9 (14) | 10 (16) | NR | Low |
| Xu W, 2021 [130] | Retrospective cohort | China | ICU and non-ICU | 659 | Adults/Children | 50.4 | 5.0 | NR | NR | NR | 48 (7) | NR | Low |
| Yao T, 2020 [131] | Retrospective cohort | China | NR | 83 | Adults | 63.9 | | 71 (86) | 83 (100) | NR | 36 (43) | NR | Low |
| Yu C, 2020 [132] | Retrospective cohort | China | NR | 128 | Adults | 43.0 | | NR | 14 (11) | 64 (50) | 5 (4) | NR | Low |
| Yue H, 2020 [133] | Retrospective cohort | China | NR | 307 | Adults | 47.3 | | NR | NR | 176 (57) | NR | NR | Medium |
| Yusuf E, 2021 [134] | Case-control | Netherlands | ICU | 92 | Adults | 76.1 | 100.0 | NR | NR | NR | NR | 10 (11) | High |
| Zhang C, 2020 [135] | Retrospective cohort | China | NR | 34 | Pediatric | 41.0 | | NR | NR | 13 (38) | 9 (26) | NR | Low |
| Zhang H, 2020 [136] | Retrospective cohort | China | NR | 38 | Adults | 84.2 | | 23 (61) | 8 (21) | NR | 37 (97) | 3 (8) | Low |

[a]ICU: intensive care unit.

[b]NR: Not reported.

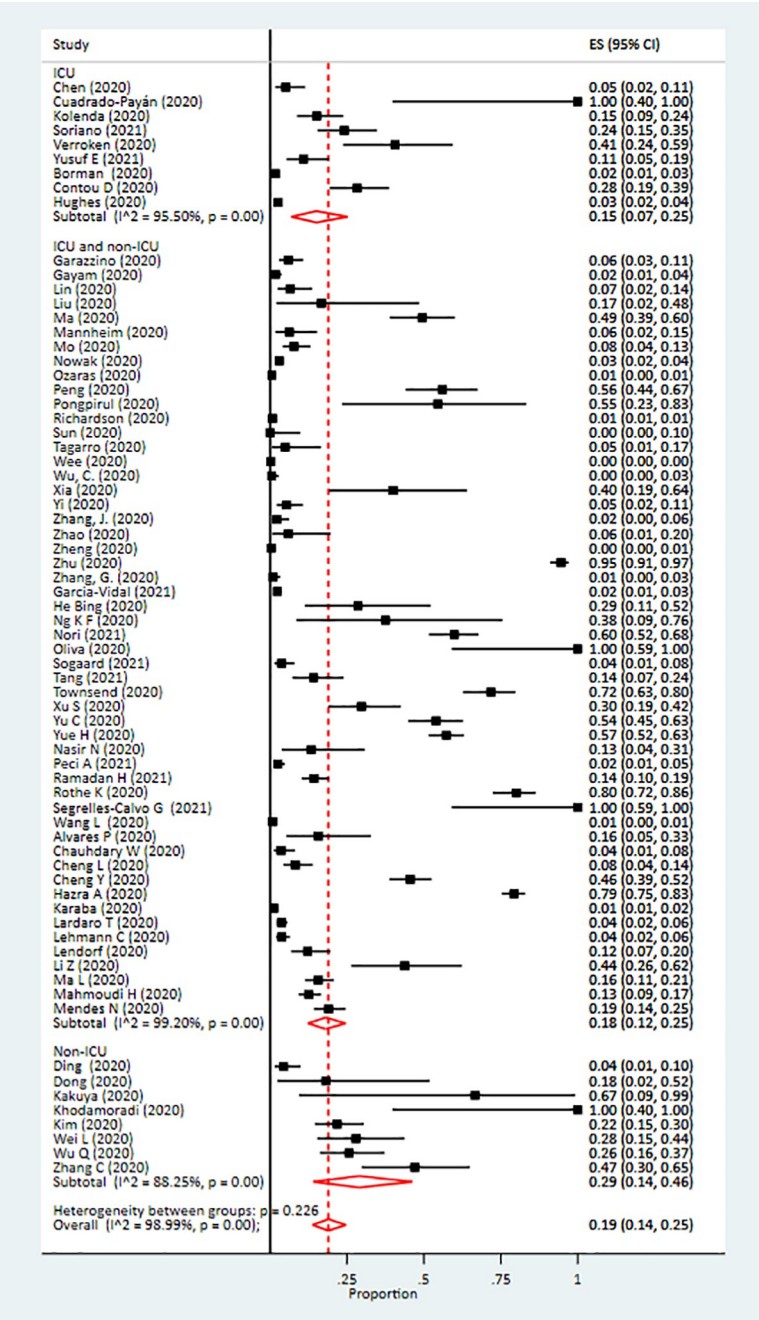

**Fig 2. Forest plot of pooled prevalence of co-infection in patients infected with SARS-CoV-2.**

The overall prevalence of comorbidities was 42% (95% CI: 35%-49%). Among those with co-infections, the prevalence of comorbidities was 32% (95% CI: 24%-41%), while it was 54% (95% CI: 42%-65%) among those who were super-infected.

Patients with a co-infection or superinfection had a higher odds of dying than those who only had SARS-CoV-2 infection (odds ratio [OR] = 3.31, 95% CI: 1.82–5.99). Subgroup analysis of mortality showed similar results, where the odds of death was higher among patients who were co-infected (OR = 2.84; 95% CI: 1.42–5.66) and those who were super-infected

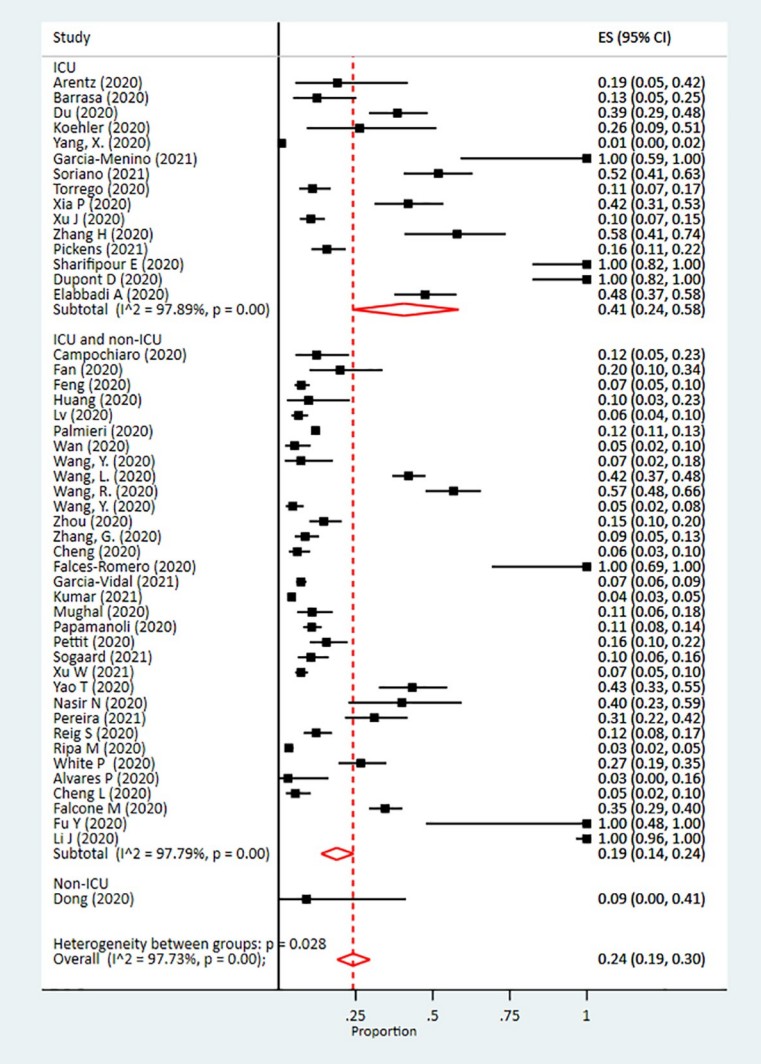

**Fig 3. Forest plot of pooled prevalence of superinfection in patients infected with SARS-CoV-2.**

(OR = 3.54; 95% CI: 1.46–8.58). There was a higher prevalence of mechanical ventilation among patients with superinfections (45% [95% CI: 33%-58%]) compared to those with co-infections (10% [95% CI: 5%-16%]). Fifty studies reported data on average LOS. The average LOS for co-infected patients was 29 days (standard deviation [SD] = 6.7), while the average LOS for super-infected patients was 16 days (SD = 6.2). None of the studies included in this meta-analysis reported data on discharge disposition and readmissions.

## Risk of bias assessment

Sixty-two percent (73/118) of studies were rated as having low risk of bias, 34% (40/118) as having medium risk of bias, and 4% (5/118) as having a high risk of bias.

## Discussion

We found that 19% of patients with SARS-CoV-2 were co-infected with other pathogens, and the prevalence of co-infection was higher among patients who were not in the ICU (29%). We

**Table 2. All identified organisms as a proportion of total number of organisms per pathogen.**

| Pathogen type | Co-infection (N = 1910) No. (%) | Superinfection (N = 480) No. (%) |
|---|---|---|
| **Bacteria** | | |
| *Staphylococcus aureus* | 148 (7.7) | 13 (2.7) |
| *Haemophilus influenza* | 127 (6.6) | 6 (1.3) |
| *Mycoplasma pneumoniae* | 82 (4.3) | 6 (1.3) |
| *Acinetobacter spps* | 78 (4.1) | 107 (22.3) |
| *Escherichia coli* | 73 (3.8) | 33 (6.9) |
| *Stenotrophomonas maltophilia* | 10 (0.5) | 18 (3.8) |
| *Klebsiella pneumoniae* | 189 (9.9) | 28 (5.8) |
| *Streptococcus pneumoniae* | 156 (8.2) | 4 (0.8) |
| *Chlamydia pneumoniae* | 29 (1.5) | 0 (0) |
| *Bordetella* | 3 (0.2) | 0 (0) |
| *Moraxella catarrhalis* | 32 (1.7) | 2 (0.4) |
| *Pseudomonas* | 67 (3.5) | 52 (10.8) |
| *Enterococcus faecium* | 14 (0.7) | 22 (4.6) |
| **Viruses** | | |
| Non-SARS-CoV-2[a] coronavirus strains | 38 (2.0) | 9 (1.9) |
| Human influenza A | 426 (22.3) | 0 (0) |
| Human influenza *B* | 73 (3.8) | 0 (0) |
| Respiratory syncytial virus | 72 (3.8) | 2 (0.4) |
| Parainfluenza | 17 (0.9) | 0 (0) |
| Human metapneumovirus | 20 (1.0) | 9 (1.9) |
| Rhinovirus | 68 (3.6) | 11 (2.3) |
| Adenovirus | 35 (1.8) | 2 (0.4) |
| **Fungi** | | |
| *Mucor* | 6 (0.3) | 1 (0.2) |
| *Candida spp.* | 19 (1.0) | 90 (18.8) |
| *Aspergillus* | 128 (6.7) | 65 (13.5) |

[a]SARS-CoV-2: severe acute respiratory syndrome coronavirus 2.

also found a higher prevalence of superinfection compared to co-infection (24%), particularly among ICU patients (41%). Further, we found that super-infected patients had a higher prevalence of mechanical ventilation and comorbidities, and a higher risk of death.

Two previous reviews found a prevalence of bacterial co-infection of 7–8% and viral co-infection of 3% in SARS-CoV-2 infected patients, which are lower than our estimates [11, 12]. We extended this work by distinguishing between super- and co-infection because of the different implications of co-infections vs. superinfections. In particular, bacteria and other pathogens have been shown to complicate viral pneumonia and lead to poor outcomes [137]. In addition, our review spanned a longer period of time and included many newer studies, which may further account for differences in prevalence data.

The three most frequently identified bacteria among co-infected patients in our study were *Klebsiella pneumonia*, *Streptococcus pneumoniae*, and *Staphylococcus aureus*. *Streptococcus pneumoniae* is a frequent cause of superinfection in other respiratory infections, such as influenza [138]. A study by Zhu et al. showed similar results [67], and a review by Lansbury et al. showed that *Klebsiella pneumoniae* and *Haemophilus influenza* were some of the most frequent bacterial co-infecting pathogens [11]. As expected, *Staphylococcus aureus* also was present in a

sizeable number of cases. The most frequent bacteria identified in super-infected patients was *Acinetobacter spp.*, which is a common infection, especially in ventilated patients [139].

In our study, the three most frequently identified viruses among co-infected patients were influenza type A, influenza type B, and respiratory syncytial virus. These findings are important particularly for influenza because testing constraints continue to exist, yet clinical presentation of influenza and SARS-CoV-2 is similar. There are major infection control and clinical implications of missing a SARS-CoV-2 or influenza diagnosis if co-infection is not considered and diagnostic testing for both pathogens is not undertaken.

Our findings have implications for infection preventionists, clinicians, and laboratory leaders. Respiratory virus diagnostic testing protocols should take into account that co-infection with SARS-CoV-2 is not infrequent, and therefore viral panel testing may be advisable in patients with compatible symptoms. Treatment protocols should also include assessment for co-infections, particularly influenza, so that appropriate treatment for both SARS-CoV-2 and influenza can be administered.

Another key finding from our study was that co-infection or superinfection was associated with an increased odds of death. This is consistent with other studies that have shown a positive association between co-infection or superinfection and increased risk of death among patients with the SARS-CoV-2 infection [140, 141].

Our study showed that antibiotics were administered in 98% of the 83 studies that reported this data. The type of antibiotics (i.e., broad or narrow spectrum) were not widely ascertainable, as these details were not provided in many studies. In the spirit of antibiotic stewardship, antibiotic use even in SARS-CoV-2 infected patients should be judicious and only in cases with an objective diagnosis of bacterial co-infection.

Our study has limitations. We were not able to assess important outcomes, such as discharge disposition and hospital readmissions, due to a lack of these data in the included studies. We were also not able to document time to superinfection, as the included studies did not report this information. Studies provided the number of patients with superinfections without stating the exact time when this determination was made after SARS-CoV-2 diagnosis. Most of the studies included in the meta-analysis were case series with their inherent limitations [142]. It is possible that some of the pathogens that were reported as superinfections or secondary infections were present but not tested for at admission and hence were co-infections. It was not possible to assess this from the studies. There was significant heterogeneity in the studies, as was anticipated given the variation in settings, patient populations, and diagnostic testing platforms across the studies.

## Conclusions

Our study showed that as many as 19% of patients with COVID-19 have co-infections and 24% have superinfections. The presence of either co-infection or superinfection was associated with poor outcomes, such as increased risk of mortality. Our findings support the need for diagnostic testing to identify and treat co-occurring respiratory infections among patients with SARS-CoV-2 infection.

## Supporting information

**S1 Fig. Forest plot of pooled prevalence of viral respiratory co-infections and viral superinfections in patients infected with SARS-CoV-2.**
(TIF)

**S2 Fig. Forest plot of pooled prevalence of bacterial co-infections and bacterial superinfections in patients infected with SARS-CoV-2.**
(TIF)

**S3 Fig. Forest plot of pooled prevalence of fungal co-infections and fungal superinfections in patients infected with SARS-CoV-2.**
(TIF)

**S1 File. Study protocol.**
(PDF)

**S2 File. Supplementary material: Search strategies, COVID-19 and co-infections, and final search.**
(PDF)

**S3 File. PRISMA 2009 checklist.**
(PDF)

**S4 File. Data used for the analysis.**
(XLSX)

## Author Contributions

**Conceptualization:** Jackson S. Musuuza, Nasia Safdar.

**Formal analysis:** Jackson S. Musuuza, Lauren Watson, Vishala Parmasad, Nasia Safdar.

**Funding acquisition:** Nasia Safdar.

**Methodology:** Jackson S. Musuuza, Lauren Watson, Vishala Parmasad, Nathan Putman-Buehler, Leslie Christensen.

**Project administration:** Jackson S. Musuuza.

**Software:** Leslie Christensen.

**Supervision:** Nasia Safdar.

**Writing – original draft:** Jackson S. Musuuza, Nasia Safdar.

**Writing – review & editing:** Jackson S. Musuuza, Lauren Watson, Vishala Parmasad, Nathan Putman-Buehler, Leslie Christensen, Nasia Safdar.

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
