## [Decision Letter · Decision Letter 0]

22 Jan 2021

PONE-D-20-33286

Prevalence and outcomes of co-infection and super-infection with SARS-CoV-2 and other pathogens: A Systematic Review and Meta-analysis

PLOS ONE

Dear Dr. Musuuza,

Thank you for submitting your manuscript to PLOS ONE. After careful consideration, we feel that it has merit but does not fully meet PLOS ONE’s publication criteria as it currently stands. Therefore, we invite you to submit a revised version of the manuscript that addresses the points raised during the review process.

During the revision process, please address the comments related to discussion of the findings in the context of the recent understanding of co- and super-infections with SARS-CoV-2.

We look forward to receiving your revised manuscript.

Kind regards,

Victor C Huber

Academic Editor

PLOS ONE

Journal Requirements:

3. We note that this manuscript is a systematic review or meta-analysis; our author guidelines therefore require that you use PRISMA guidance to help improve reporting quality of this type of study. Please upload copies of the completed PRISMA checklist as Supporting Information with a file name “PRISMA checklist”.

Reviewers' comments:

Reviewer's Responses to Questions

**Comments to the Author**

1. Is the manuscript technically sound, and do the data support the conclusions?

Reviewer #1: Partly

Reviewer #2: No

2. Has the statistical analysis been performed appropriately and rigorously? 

Reviewer #1: I Don't Know

Reviewer #2: Yes

3. Have the authors made all data underlying the findings in their manuscript fully available?

Reviewer #1: Yes

Reviewer #2: Yes

4. Is the manuscript presented in an intelligible fashion and written in standard English?

Reviewer #1: Yes

Reviewer #2: Yes

5. Review Comments to the Author

Reviewer #1: The article by Musuuza et al. investigates the prevalence and outcomes of co/ super-infection with SARS-CoV-2 as A Systematic Review and Meta-analysis.

It is an interesting study and definitely important to bring attention to other infections among COVID-19 patients.

Major comments

-Due to the importance of the disease, the evaluation period of the articles is very short and many interesting and newly published articles have been ignored. For example (https://pubmed.ncbi.nlm.nih.gov/32873235/, https://pubmed.ncbi.nlm.nih.gov/32613024/ , https://pubmed.ncbi.nlm.nih.gov/32603803/ , etc).

It is suggested that the author increase the time period for reviewing articles and add newer studies to the MS.

- No data on the use of antibiotics in SARS-CoV-2 patients were found in this study. It is recommended to add some data about the treatment protocols used in patients.

Discussion

The results are not well discussed, especially the role of co/super infections in mortality of COVID patients. So, it needs to be improved.

Conclusion

The sentence " Our results have ………. virus season in the fall." cannot be concluded from this study.

Major comment

Method

Page 4, Line 83 - change "Covid" to "COVID".

Result

In Table 2, change "Fungus" to "Fungi".

Reviewer #2: This is well drafted manuscript on a very relevant question. Authors have done adequate work although due to dynamic nature of the ongoing pandemic the findings may vary in near future and further updates on this issue will be useful.

6. PLOS authors have the option to publish the peer review history of their article (what does this mean?). If published, this will include your full peer review and any attached files.

Reviewer #1: No

Reviewer #2: No

---

## [Author Response · Author response to Decision Letter 0]

14 Apr 2021

April 14, 2021

RE: PONE-D-20-33286: “Prevalence and outcomes of co-infection and super-infection with SARS-CoV-2 and other pathogens: A Systematic Review and Meta-analysis.” 

Dear Dr. Huber,

We thank you and the reviewers for the careful review and thoughtful feedback on our manuscript, “Prevalence and outcomes of co-infection and super-infection with SARS-CoV-2 and other pathogens: A systematic review and meta-analysis.” We have revised the manuscript according to the comments and believe that it is substantially improved with the incorporation of these edits. 

Below, we provide a point-by-point reply to the reviewers’ comments. We have included a marked copy of the revised manuscript that highlights changes, as well as a clean version. We have also ensured that our manuscript meets style requirements of PLOS ONE.

Thank you for your consideration of our revised manuscript. 

EDITOR COMMENTS 

Comment 1: During the revision process, please address the comments related to discussion of the findings in the context of the recent understanding of co- and super-infections with SARS-CoV-2.

Authors’ reply: We have revised the Discussion to place our findings in the context of the recent understanding of co- and super-infections with SARS-CoV-2.

Comment 2: We note that you have indicated that data from this study are available upon request. PLOS only allows data to be available upon request if there are legal or ethical restrictions on sharing data publicly. For information on unacceptable data access restrictions, please see http://journals.plos.org/plosone/s/data-availability#loc-unacceptable-data-access-restrictions.

a) If there are set ethical or legal restrictions on sharing a de-identified data, please explain them in detail (e.g., data contain potentially identifying or sensitive patient information) and who has imposed them (e.g., an ethics committee). Please also provide contact information for a data access committee, ethics committee, or other institutional body to which data requests may be sent.

Authors’ reply: We have uploaded an anonymized dataset as one of the supporting information files. There are no ethical or legal restrictions on sharing our data. 

Comment 3: We note that this manuscript is a systematic review or meta-analysis; our author guidelines therefore require that you use PRISMA guidance to help improve reporting quality of this type of study. Please upload copies of the completed PRISMA checklist as Supporting Information with a file name “PRISMA checklist”.

Authors’ reply: We have included a completed PRISMA checklist as a supporting information file (S3 File).

REVIEWER #1 COMMENTS

The article by Musuuza et al. investigates the prevalence and outcomes of co/ super-infection with SARS-CoV-2 as A Systematic Review and Meta-analysis.

It is an interesting study and definitely important to bring attention to other infections among COVID-19 patients.

Major comments

Comment #1: Due to the importance of the disease, the evaluation period of the articles is very short and many interesting and newly published articles have been ignored. For example (https://pubmed.ncbi.nlm.nih.gov/32873235/, https://pubmed.ncbi.nlm.nih.gov/32613024/ , https://pubmed.ncbi.nlm.nih.gov/32603803/ , etc.).

It is suggested that the author increase the time period for reviewing articles and add newer studies to the MS.

Authors’ reply: As suggested, we have expanded the timeframe for the search to include eligible articles published since our last search date (June 11, 2020) through February 8, 2021.

Comment #2: No data on the use of antibiotics in SARS-CoV-2 patients were found in this study. It is recommended to add some data about the treatment protocols used in patients.

Authors’ reply: Seventy percent (83/118) of the studies reported data on antibiotic use. Of these, antibiotics were administered in 98% (81/83) of the studies. We have included this information in the revision.

Discussion

Comment #3: The results are not well discussed, especially the role of co/super-infections in mortality of COVID patients. So, it needs to be improved.

Authors’ reply: We have revised the Discussion overall and included a paragraph on the role of co/super-infections in mortality of SARS-COV-2 infected patients. We believe the discussion is much improved with this revision.

Conclusion

Comment #4: The sentence " Our results have ………. virus season in the fall." cannot be concluded from this study.

Authors’ reply: We agree with the reviewer and have removed this sentence and revised the Conclusion accordingly.

Methods

Comment #5: Page 4, Line 83 - change "Covid" to "COVID".

Authors’ reply: We thank the reviewer for this comment, and we would like to clarify that the term “Covid” was used here as a search term since there some studies have used it in their reports. Throughout the paper, we use “COVID-19.” 

Results

Comment #6: In Table 2, change "Fungus" to "Fungi".

Authors’ reply: We have made this correction per the reviewer’s suggestion.

REVIEWER #2 COMMENTS

Reviewer #2: This is well drafted manuscript on a very relevant question. Authors have done adequate work although due to dynamic nature of the ongoing pandemic the findings may vary in near future and further updates on this issue will be useful.

Authors’ reply: We thank the reviewer for this comment. Although, we have extended our article search dates in this revision, we agree that further updates of this work will be needed periodically.

---

## [Editor Report · Decision Letter 1]

22 Apr 2021

Prevalence and outcomes of co-infection and superinfection with SARS-CoV-2 and other pathogens: A systematic review and meta-analysis

PONE-D-20-33286R1

Dear Dr. Musuuza,

We’re pleased to inform you that your manuscript has been judged scientifically suitable for publication and will be formally accepted for publication once it meets all outstanding technical requirements.

Kind regards,

Victor C Huber

Academic Editor

PLOS ONE
---

## [Editor Report · Acceptance letter]

28 Apr 2021

PONE-D-20-33286R1 

Prevalence and outcomes of co-infection and superinfection with SARS-CoV-2 and other pathogens: A systematic review and meta-analysis 

Dear Dr. Safdar:

I'm pleased to inform you that your manuscript has been deemed suitable for publication in PLOS ONE. Congratulations! Your manuscript is now with our production department. 

Kind regards, 

on behalf of

Dr. Victor C Huber 

Academic Editor

PLOS ONE